# Influence of Dose Conversions, Equilibrium Factors, and Unattached Fractions on Radon Risk Assessment in Operating and Show Underground Mines

**DOI:** 10.3390/ijerph20085482

**Published:** 2023-04-12

**Authors:** Krystian Skubacz, Katarzyna Wołoszczuk, Agata Grygier, Krzysztof Samolej

**Affiliations:** 1Central Mining Institute, Silesian Centre for Environmental Radioactivity, Plac Gwarków 1, 40-166 Katowice, Poland; agrygier@gig.eu (A.G.); ksamolej@gig.eu (K.S.); 2Central Laboratory for Radiological Protection, ul. Konwaliowa 7, 03-194 Warszawa, Poland; woloszczuk@clor.waw.pl

**Keywords:** radon hazards, underground active and show mines, cave, evaluation of dose conversion, unattached fraction, equilibrium factors, assessment of effective dose

## Abstract

This paper compares the results of measurements taken in the underground workings of active and tourist mines. In these facilities, the aerosol size distributions of ambient aerosols at key workplaces and the distributions of radioactive aerosols containing radon decay products were determined. Based on these studies, dose conversions used for dose assessment and unattached fractions were determined. In addition, radon activity concentrations and potential alpha energy concentrations of short-lived progeny were also measured in the mines to determine the equilibrium factor. The dose conversions varied between 2–7 mSv/(mJ × h × m^−3^). The unattached fraction measured in active coal mines ranged from 0.01–0.23, in tourist mines from 0.09–0.44, and in the tourist cave it was 0.43. The results showed significant discrepancies between the effective doses determined from current recommendations and legal regulations and those determined from direct measurements of parameters affecting exposure.

## 1. Introduction

The occurrence of increased natural radioactivity in Polish coal mines was discovered in the 1960s [1]. In the 1970s, it was identified that the source of enhanced gamma radiation could be sediments formed by the precipitation of radium isotopes from formation waters flowing into the underground workings from the rock mass [2]. Further studies conducted in later years on the phenomenon of natural radioactivity in coal mines helped identify additional sources of natural ionising radiation in these mines. It turned out that the primary sources of natural ionising radiation in mines are waters containing radium isotopes, precipitates from these waters, and short-lived radon decay products [3,4,5,6,7]. These problems are not only the domain of Polish coal mines. Radium-bearing waters occur in both gas- and oil-bearing deposits [8] and deep-sea waters [9], while reports and publications describing the occurrence of similar phenomena in German coal mines appeared in the late 1970s and early 1980s [10].

Hazard assessment of natural radionuclides is mandatory in Polish underground mines and covers all identified sources of natural radioactivity. Long-term radiometric monitoring has revealed that radon (Rn-222) decay products contribute the most to miners’ exposure doses [11]. Measurements were made intermittently to detect the presence of thorium decay products from the thorium series. However, their concentrations were always negligibly small.

Contemporary epidemiological studies indicate that radon is responsible for inducing 3–14% of lung cancers. The risk increases significantly with prolonged exposure to a radon activity concentration of 100 Bq/m^3^ and progresses by 16% per 100 Bq/m^3^ [12]. Concentrations at this level are quite common in living rooms and given the human lifestyle, the exposure can be classified as long-term. The US Environmental Protection Agency (EPA) has concluded that the presence of radon in residential buildings is the second leading cause of lung cancer induction, after cigarette smoking [13]. The radon risk can increase significantly in cases of poorer ventilation, such as in underground workplaces.

The importance of these risks has also been recognised in the European Union, where a Directive enacted in 2013 devotes a large section to radon risk issues in residential buildings and workplaces [14]. Similarly, in other countries, the control of radon risk is the subject of legal requirements for residential buildings and workplaces located above- and underground [15,16,17,18,19].

Although various formal documents, including legislation, refer in simplified terms to ‘radon risk’, the EU Directive makes it clear that this means exposure to “radon (Rn-222) and its short-lived decay products” (Po-218, Pb-214, Bi-214, and Po-214) or thoron (Rn-220) and its decay products (Po-216, Pb-212, Bi-212, and Po-212), which belong to the uranium and thorium series, respectively. Unlike radon in the gaseous form, its decay products attach in the first phase to gases or water vapour molecules present in the air to form clusters. In the next stage, the clusters or free atoms, the so-called unattached fraction, form larger complexes with ambient aerosols, the so-called attached fraction. Unlike radon, which is largely removed with exhaled air, radioactive aerosols are deposited in the respiratory system, and deposition efficiency depends on their size. As a result, the contribution of short-lived radon progeny to the total dose in comparison to radon is predominant, reaching 98% for aerosol sizes of 250 nm [20], while only 2% is the result of radon ingress alone. Similar proportions are also indicated by the recent ICRP report [21].

A measure of the hazard from radionuclides is the effective dose, which in the case of short-lived radon or thoron decay products is calculated based on the following formula:(1)$$E=kC_{\alpha }t$$
where

C_α_ is the potential alpha energy concentration (PAEC) defined as the ratio of the total energy of alpha particles released by short-lived radon progeny (Po-218, Pb-214, Bi-214, and Po-214) or thoron progeny (Po-216, Pb-212, Bi-212, and Po-212), respectively, as a result of their complete radioactive decay per unit volume of air in which these nuclides are present, expressed in [J × m^−3^]; k is the dose conversion, defined as the ratio of the dose expressed in sieverts to the effective dose units per exposure, which is the product of the potential alpha energy concentration and the exposure time, expressed in [Sv/(J × h × m^−3^)]; and **t** is the exposure time, in hours.

The dose conversion k is a function of the radioactive aerosol size distribution, which depends on the concentration and size distribution of the ambient aerosols, air pressure, and temperature. This is because the physical properties of such aerosols determine with what efficiency and at what location in the respiratory system they are deposited. Particularly important is the unattached fraction, which is expressed by the ratio of the PAEC corresponding to this class of aerosols and the total PAEC. This parameter is very important because the dose conversions are particularly high for particles associated with the unattached fraction, which consists of nanometre-sized objects with an average size of 1.2 nm [22]. The diffusion coefficient is relatively high for small objects, resulting in increased deposition in the respiratory system and consequently increasing the effective dose [23].

In Poland, issues related to radiological protection in underground mines are included in the general system of radiological protection. The basic principles, such as exposure limits, reference levels, classification of workers and workplaces, etc., are regulated by the Atomic Law and relevant executive legal regulations. As natural radioactivity is identified as one of the natural hazards occurring in underground mines, the treatment of radiation exposure is also regulated by the Geological and Mining Law. Such a solution makes it possible to adapt the general principles derived from the Atomic Law to the specific conditions found in underground mining facilities. Polish regulations stipulate the following dose conversions for assessing the effective dose based on PAEC measurement: 1.4 mSv/(mJ × h × m^−3^) for radon and 0.5 mSv/(mJ × h × m^−3^) for thoron. Exceeding an annual dose of 1 mSv triggers notification under the procedure outlined in the Directive [14]. Subsequent actions are subject to the decision of the relevant supervisory authority.

The work includes the results of measurements carried out in active and tourist mines. The studies involve determining the size distributions of ambient and radioactive aerosols in relation to radon decay products. These determinations provide a basis for assessing the dose conversions. In these facilities, the unattached fraction, the radon activity concentration, and the PAEC of radon decay products were also determined and used to estimate doses according to current regulations. Since the effective doses associated with the presence of radon decay products can also be estimated based on the radon activity concentration (this is a commonly used method), measurements of radon activity concentration were also performed.

The method requires adopting or determining the equilibrium factor F, which is defined as the ratio of the equilibrium equivalent activity concentration of radon (EEC) to the actual radon activity concentration. The equilibrium equivalent concentration is the activity concentration of radon (or thoron) in radioactive equilibrium with its short-lived progeny (Thoron progeny) that would have the same potential alpha energy concentration as the actual (nonequilibrium) mixture. The equilibrium factor can be calculated based on the semi-empirical formula:(2)$$F=\frac{C_{\alpha }}{5.6\times 10^{-3}C_{Rn}}$$
where the potential alpha energy concentration C_α_ is expressed in (μJ/m^3^) and the radon activity concentration C_Rn_ in (Bq/m^3^).

When ventilation is poor, F values are typically close to unity, but they decrease as radioactive aerosols are removed more effectively. Typically, a specifically recommended equilibrium factor is adopted to calculate doses based on measurements of radon activity concentration. Therefore, the assessment of doses in such a case may be subject to a large uncertainty because the real value of the equilibrium factor can vary over a wide range, especially in facilities with a mechanical ventilation system. That is why the reference levels in underground mines are often expressed not as radon activity concentrations but directly as the PAEC [19]. This approach has also been adopted in Polish regulations.

Dose estimates based on thoron activity concentration are not provided at all for the thoron hazard. While dose conversions for calculating dose based on radon activity concentration can be found in a recent report of the International Commission on Radiological Protection [21], for thoron only their values related to the PAEC or equilibrium equivalent concentration of thoron decay products are provided.

## 2. Materials and Methods

Measurements made in underground workings cover determinations of radon activity concentrations and the PAEC of radon decay products, as well as measurements of the unattached fraction and ambient and radioactive aerosol size distributions. The following equipment was used during their implementation:AP-2000EX aspirators (Two-Met, Zgierz, Poland) equipped with systems for separating the respirable fraction and alpha probes containing thermoluminescence detectors made of CaSO_4_:Dy phosphor. During measurements, the air is pumped through a filter located above the alpha probe, which contains three measuring heads, each with two thermoluminescent detectors inside. One of the detectors, separated by a spacer, is intended for background measurement. The aspirators are powered by rechargeable batteries and can run for up to eight hours. After exposure, the thermoluminescent detectors are read out using dedicated thermoluminescent readers to assess the alpha potential energy concentration [24].The AlphaGuard-EF radiometer (Genitron, Frankfurt, Germany; Bertin Technologies, Montigny-le-Bretonneux, France) is designed for long-term measurements of radon activity concentration in air. In the diffusion mode, radon enters by diffusion in a 0.56 L ionisation chamber operating at 750 V. A glass filter prevents impurities and radon decay products from infiltrating the chamber. As a result, only radon is present inside the chamber initially. During diffusion mode, the analysis of concentrations can be performed at one of two user-selectable frequencies: every ten or every sixty minutes. However, air movement can also be forced, resulting in a faster response of the measurement system to external conditions. During this mode, the analysis can take place every one minute or every ten minutes. The analysis of the generated pulses includes both their height and shape. The device can operate in a range of up to several tens of kilobecquerels per cubic meter. Simultaneously, the device can also record humidity, pressure, and air temperature.Solid-state nuclear track detectors (Radosys Group, Budapest, Hungary). Radon activity concentrations in the air were measured using trace detectors equipped with CR-39 film, which is a plastic polymer of allyl diglycol carbonate that is sensitive to alpha particles emitted by radon and its decay products. These alpha particles cause microscopic defects in the polymer structure, referred to as traces, by breaking down the chemical bonds in the material. After exposure, the detectors are etched using 25% NaOH, and the number of traces is counted using an optical microscope. The radon activity concentration is directly proportional to the number of traces on the CR-39 film. Typically, the detectors are exposed for a period of 1–3 months, and this method can measure radon activity concentrations as low as 10 Bq/m^3^ if the measurement period is three months.Lucas chamber with scintillation detector from SARAD GmbH (Dresden, Germany). The measurement consists of taking an air sample in the chamber, which is coated inside with alpha-sensitive ZnS(Ag). Then, the generated light is detected by a photomultiplier.Particle spectrometers for determining size distributions of ambient aerosols. The study employed two spectrometers manufactured by TSI Incorporated (USA, Shoreview, MN): the SMPS (Scanning Mobility Particle Sizer) spectrometer and the APS (Aerodynamic Particle Sizer) spectrometer. The former is specifically designed to determine the size distribution of smaller aerosol particles. It is equipped with an electrostatic classifier with a particle mobility analyser and a particle counter. Depending on the type of particle mobility analyser (DMA), the measuring range can start from 3–4 nm (Nano DMA) or several nanometres (Long DMA). An impactor is placed at the air inlet of the measurement system to avoid interference from larger particles that would fall outside the measurement range of the spectrometer. After passing the impactor, the air is directed into a column containing a Kr-85 beta radiation source, which ionises the air. As a result, the aerosol particles become charged and reach a state of electrostatic equilibrium. The differential analyser, which the air stream reaches next, contains a negatively charged electrode in the central part. As a result, the trajectories of the positively charged aerosol particles in the laminar air stream curve towards the electrode. The particle trajectories depend on both the particles’ voltage and mobility (size). As a result, the particles arrive sequentially at the aperture at the bottom of the DMA, from the smallest to the largest size, before being directed to a particle counter equipped with a laser system. The APS spectrometer is a less complex device always operating in the aerodynamic particle size range from 0.5 µm to 20 µm, divided into 52 channels. Just behind the inlet to the device, the air stream is split at a ratio of 4:1. The larger air volume is filtered and combined with the stream containing aerosols near the laser system. As a result, the particles are subject to acceleration and the flight time between the two laser beams depends on their size. Finally, their size and intensity are evaluated after comparison with the calibration curve.The Radon Progeny Particle Size Spectrometer (RPPSS) can measure the particle size distribution of short-lived radon progeny in the diameter range from 0.6 nm to 2494 nm and determine the dose conversions. It is a research instrument, designed and constructed by the Australian Radiation Protection and Nuclear Safety Agency (ARPANSA, Yallambie, VIC 3085, Australia). The RPPSS spectrometer has eight parallel measurement stages, which employ four diffusion screens connected in parallel and three single-stage impactors for particle size selection. In addition, one of the components of the system is an open filter that captures particles without any selection, which, in combination with the results obtained for the other tracks, allows to determine the unattached fraction of the radon progeny. Filters and mylar films are located behind diffusion grids and impactors, respectively. The instrument uses solid-state surface barrier silicon detectors operating in a spectrometric system to measure the activity of radionuclides deposited on the filters and mylar films. The potential alpha energy values obtained from each stage provide input to two independent deconvolution algorithms, Twomey [25] and EMax (Expectation Maximisation) [26], which yield concentration distributions of the potential alpha energy for each stage and the particle distribution of the short-lived radon decay products. The process of finding the best fit of the **f** distribution to the measurement results M follows the following fitting procedures:

Twomey algorithm
(3)$$f_{j}^{\left(r\right)}=f_{j}^{\left(r-1\right)}\left(1+\left(\frac{M_{i}}{\sum_{j}T_{ij}f_{j}^{\left(r-1\right)}}-1\right)T_{ij}\right)$$

Emax algorithm
(4)$$f_{j}^{\left(r\right)}=f_{j}^{\left(r-1\right)}\frac{\sum_{i}T_{ij}M_{i}}{\sum_{j}T_{ij}f_{j}^{\left(r-1\right)}}\frac{1}{\sum_{i}T_{ij}}$$
where f_j_ is the activity size distribution of particles of the *j*-th size, M_i_ represents the measurement result corresponding to particles, that make it through the *i*-th stage, T_ij_ the fractional penetration of the *j*-th size particle through the *i*-th stage, and r is the iteration number.

The algorithms were developed in the 1970s and 1980s and are still frequently used to reconstruct size distributions of radioactive aerosols using data from diffusion screens or impactors. The Twomey and Emax algorithms lead to similar solutions. Fewer iterations are performed in Twomey than in the EMax algorithm because the number of updates per iteration is greater by a factor equal to the number of measurement stages. This is offset by the greater stability of the EMax algorithm due to the averaging of the entire dataset in each iteration.
The RGR-40 mining radiometer is designed for rapid measurements of potential alpha energy concentrations. The instrument is equipped with a membrane pump, which pumps air for 5 min through a glass fibre filter on which radon decay products are collected. A silicon detector then records the alpha radiation emitted by these nuclides, and the entire measurement cycle is completed within 15 min.

## 3. Results

### 3.1. Measurement Sites

Measurements were made in underground and tourist underground mines and in a cave open to the public. Tourist facilities were located in Lower and Upper Silesia, while active hard coal mines in Upper Silesia in Poland

The use of measurement equipment is limited in underground mines due to methane hazards, general safety rules, and severe environmental conditions. As a result, the planned set of measurements was not carried out at all measurement sites; this applies especially to measurements carried out with the RPPSS spectrometer for the determination of radioactive aerosol size distributions.

Measurements of the ambient aerosol size distributions were conducted in all key underground mine workings (Figure 1) of an active coal mine: in the airflow coming in from the surface to ventilate the mine workings (downcast shaft), in the corridor workings to make the deposit available (cross-cut), in the area where underground water is collected before being pumped to the surface (water gallery), in the area where the corridor workings are being excavated (road head), in the area where coal extraction was currently taking place (coal face, longwall), and in the airflow discharged from the mine to the surface after ventilating the mine workings (upcast shaft). At times, the environmental conditions became highly unfavourable because of the intense dust generated in the vicinity of the mining operations (Figure 2). The unattached fraction was estimated using the semi-empirical Formula (2) based on the total aerosol concentration.

Similar measurements were conducted at four tourist mines, where metal or uranium ores were previously extracted. Mining operations have been discontinued due to the depletion of the deposits, and the sites have been made accessible to the public. Compared to active coal mines, where exploitation is carried out at a depth of around 1000 m, these mines are relatively shallow, located a few tens of meters underground, or in adits drilled into the rock mass from the mountainside into the deposit. Compared to active mines, there are no heavily dusty areas (Figure 3). However, while mechanical ventilation is present in active mines, particularly in areas with methane hazards, these tourist mines usually rely on gravity or low-capacity mechanical ventilation. In order to gather additional data for comparison, similar measurements were taken at a publicly accessible cave, where the aerosol concentrations were recorded at very low levels (Figure 4).

### 3.2. Size Distribution of Ambient Aerosols

The ambient aerosol measurements aimed to estimate the size distributions of radioactive aerosols and, subsequently, calculate the dose conversions for radon decay products based on these measurements. The values of these coefficients were determined according to the method described in the publication [27]. The characteristics of the obtained ambient aerosol size distributions are presented in Table 1 and Table 2 and Figure 5 and Figure 6. Only areas near intense mining activities, such as the road head or coal face, showed a significant contribution of large particles.

The results of analogous measurements made in four tourist mines, and in a cave open to the public, are presented in Table 3 and Table 4 and Figure 6. Compared to the active mines, the objects open to the public are characterised by relatively short galleries, and the tours usually last about one hour. There is also a lack of operating mining equipment or areas designated for specific tasks (workplaces). Therefore, one-day surveys were carried out at one selected site taken as representative of the whole object.

After analysing the results obtained in an active coal mine, high concentrations of aerosols were found in the air stream entering the underground workings through the downcast shaft. The air supplied from the surface travelled about 1000 m, becoming enriched in aerosols as it moved through the shaft, especially due to the high humidity of the air in this space. High concentrations were observed in areas close to the mining operations during the operation of mining machines, where the contribution of coarse (2.5–10 μm) and very coarse (>10 μm) particles was noticeably higher than in other areas [27,28].

Compared to the active mine, the concentrations at the other sites are noticeably lower, by an order or even two orders of magnitude (Figure 7). For comparison, Table 3 and Table 4 present measurements taken in the tourist cave, which indicate a highly clean environment with aerosol concentrations that can be one to three orders of magnitude lower than in the other studied sites.

### 3.3. Dose Coefficients and Unattached Fraction

Table 5 shows the values of the dose conversions evaluated for the active mine under the assumption that the unattached fraction is not present and based on their estimated contribution. The coefficients were evaluated for breathing rates of 1.2 m^3^/h through the mouth and 0.75 m^3^/h through the nose [21]. The unattached fraction was estimated from the total particle concentration Z and the following relationship [28]:(5)$$f_{p}=\frac{414}{Z[{cm}^{-3}]}$$

The dose conversions are higher than the coefficient for workplaces of 1.4 mSv/(mJ × h × m^−3^), which is still in force in Polish law [29]. In comparison, the dose conversion currently recommended by the International Commission on Radiological Protection for underground mining facilities is 3 mSv/(mJ × h × m^−3^), and for indoor workplaces where workers are engaged in substantial physical activities as well as for workers in tourist caves, it is 6 mSv/(mJ × h × m^−3^), assuming an unattached fraction of 0.01 [21]. The coal mine where measurements were taken had an average estimated dose conversion of 4.8 mSv/(mJ × h × m^−3^) for the attached fraction of short-lived radon decay products, considering all measurement sites. After considering the estimated unattached fraction, the mean value of the factor is 5.2 mSv/(mJ × h × m^−3^) (Table 5). In the work of Zock [23], the average dose conversion for the attached fraction in mines is estimated to be 3.5 mSv/(mJ × h × m^−3^) if the same mode of respiration is considered [23]. The estimated values are closer to the new dose conversions recommended by the ICRP [21]. However, it is important to emphasise the high variability of the dose conversions for the different measurement sites, reaching up to almost 50% (Table 5). These differences are due to the variability in the aerosol size distribution at individual workplaces. This calls into question the possibility of using a single dose conversion for the entire mine, where the workplaces are usually permanent.

The unattached fraction used in the calculations was estimated using Relation (5) and ranged from 0.004 to 0.02, with a mean value of 0.012 ± 0.06. These values are close to the value of 0.01 used in the ICRP report [21] for determining dose conversions for a reference worker breathing at a rate of 1.2 m^3^ × h^−1^. It is important to note that Relation (5) is semi-empirical and cannot be applied to low concentrations below 4.14 × 10^8^ participles/m^3^. However, this relation provides a relatively good approximation of the actual values for active mines where aerosol concentrations are typically high.

The calculations of dose conversions performed assume that the method of determining the radioactive aerosol size distribution based on the ambient aerosol size distribution would yield appropriate results. The opportunity to verify this assumption arose during the research conducted at the tourist mines. Both the RPPSS spectrometer and the SMPS and APS ambient aerosol size distribution spectrometers were used at these facilities. Table 6 compares the values of the dose conversions determined based on both aerosols’ size distributions, with ambient aerosols measured by SMPS and APS spectrometers and radioactive ones quantified with the help of the RPPSS spectrometer (using the two calculation algorithms Twomey and Emax). The RPPSS spectrometer measured high values of unattached fractions, reaching up to 0.44 in Show mine 4 (Table 6). In the tourist cave, which is characterised by very clean air, the unattached fraction was 0.43. In contrast, the concentration of ambient aerosols in Show mine 4 was significantly lower than in the other mines but still higher than in the tourist cave (Table 3).

The dose conversions for mouth breathing at a rate of 1.2 m^3^ × h^−1^ were determined using the RPPSS spectrometer results. The values ranged from 2.1–3.1 mSv/(mJ × h × m^−3^) for the Twomey method and 2.3–3.4 mSv/(mJ × h × m^−3^) for the Emax method. Additionally, the dose conversions were calculated based on the size distributions of ambient aerosols, resulting in a range of 3.4–5.2 mSv/(mJ × h × m^−3^) (Table 6, Figure 8). The dose conversions for both mouth and nose breathing were found to be higher than the occupational exposure dose conversion of 1.4 mSv/(mJ × h × m^−3^), which is still in force in Poland and close to the currently recommended ICRP values of this parameter for mine workplaces (3 mSv/(mJ × h × m^−3^)) and in caves (6 mSv/(mJ × h × m^−3^)) [21]. For the five sites examined, the average ratio of SMPS + APS/Twomey and SMPS + APS/Emax amounts to 1.63 with a standard deviation of 0.13 and 1.54 with a standard deviation of 0.09 for mouth breathing and 1.75 with a standard deviation of 0.18 and 1.61 with a standard deviation of 0.15 for nose breathing, respectively.

### 3.4. Radon Activity Concentration, PAEC, and Equilibrium Factor

In the coal mine, the PAEC was measured using an AP-2000EX aspirator equipped with alpha probes with thermoluminescence detectors. The results are presented in Table 7. The lowest PAEC value of 0.12 µJ/m^3^ was measured at the cross-cut and the highest value of 0.66 µJ/m^3^ in the water gallery. In active mines, workers have designated workplaces, so the doses they receive can be attributed to specific positions. Assuming the current dose conversion applied in Poland and an annual working time of 1800 h, the annual effective doses would range from 0.05 to 1.41 mSv (Table 7, effective dose (1)). However, if dose conversions determined from aerosol size distributions (Table 5) and the unattached fractions are considered, the doses are much higher, ranging from 0.16 to 5.12 mSv (Table 7, effective dose (2)). It must be underlined that when calculating the effective dose based on direct PAEC measurements, there is no need to consider the equilibrium factor F.

The nature of the work carried out by tourist facility employees is different from work conducted in an active coal mine. These employees are usually guides moving along a designated route. In tourist mines and caves, there are no such restrictions as in an active mine. Therefore, both PAEC and radon activity concentrations were measured using the RPPSS and AlphaGuard spectrometers, enabling the evaluation of the equilibrium factor F (Table 8). The tested sites had varying radon activity concentrations ranging from 1050–5400 Bq/m^3^, potential alpha energy concentration within the range of 4.2–13.7 µJ/m^3^, and corresponding equilibrium factors from 0.2 to 0.8.

To determine the possible variability in the equilibrium factor in two mines, Show mine 3 and Show mine 4, radon activity concentration and PAEC were measured at several points along the tourist routes (Table 9). The radon activity concentration at tested points varied between 1060 and 4430 Bq/m^3^ in Show mine 3 and 774 and 3041 Bq/m^3^ in Show mine 4, while the PAEC varied between 1.7 and 7.6 µJ/m^3^ and 2.0 and 4.0 µJ/m^3^ in Show me 3 and in Show mine 4, respectively.

The equilibrium factor F ranges from 0.1 to 0.7, with an average of 0.4. It is estimated that the annual working time of a guide is 750 h, while the visiting time is about 1 h. The estimated doses based on the average PAEC using a dose conversion of 1.4 mSv/(mJ × h × m^−3^) would therefore be about 5 mSv for guides and 7 μSv for tourists in Show mine 3 and about 3 mSv for guides and 4 μSv for tourists in Show mine 4.

Table 10 displays the effective doses derived from measurements of PAEC and radon activity concentration, considering various dose conversions and the equilibrium factor **F**. The calculation is based on an annual working time of 1800 h. It is apparent that assuming a constant value for the equilibrium factor F in the effective dose calculation can result in either an underestimation or overestimation of the actual dose. For a dose conversion of 3 mSv/(mJ × h × m^−3^) and an equilibrium factor of 0.2, the dose values range from 4.4 to 26.6 mSv, whereas when the actual value of the equilibrium F is taken from the calculation, the dose values are in the range of 9.3–29.2 mSv. Notably, the effective dose calculations based on PAEC, radon activity concentration, and the measured equilibrium factor yield similar results. These results are over three times higher than those calculated using the radon activity concentration and the F recommended by the ICRP (Table 10). This calls into question the use of a constant equilibrium factor for the whole site, although the nature of the work carried out in tourist facilities justifies the use of an average value for the estimation of the annual effective dose.

## 4. Discussion

According to the ICRP recommendations (and general practice), the value of the equilibrium factor (F) and the corresponding dose conversion should be considered when calculating the effective dose from the radon activity concentration. The Committee’s recommendations specify fixed values for these parameters for specific facilities.

In the active mine, total aerosol concentrations were higher than in the tested cave and show mines (in the range of 1.1 × 10^10^ to 14 × 10^10^ particles/m^3^), especially in close proximity to the operating areas of the mining equipment. Only in such locations did the concentrations of coarse and very coarse particles reach 1%. Although the value of the coefficient increased for larger particles, their contribution to the total concentration was typically small. In other places, it was negligibly small. In contrast, the atmosphere in the cave had a total aerosol concentration that was several orders of magnitude lower (4.0 × 10^7^ particles/m^3^). In the tourist mines, it was in the range of 2.9 × 10^8^ – 2.3 × 10^10^ particles/m^3^.

The size distribution of aerosols affects the values of dose conversions. These were estimated using various methods. Assuming a breathing rate of 1.2 m^3^×h^−1^ through the mouth, which is typical for a standard worker [19], the coefficients varied between 2 and 7 mSv/(mJ × h × m^−3^), with an average of 4.8 mSv/(mJ × h × m^−3^) in an active coal mine, 3.0 mSv/(mJ × h × m^−3^) in tourist mines, and 3.9 mSv/(mJ × h × m^−3^) in a cave open to the public. These were always higher than the current value of 1.4 mSv/(mJ × h × m^−3^). In comparison, the current ICRP recommendations [21] suggest a dose conversion of 3.0 mSv/(mJ × h × m^−3^) for underground mining facilities and 6.0 mSv/(mJ × h × m^−3^) for caves.

The particle size distribution research found that the deposition efficiency of radioactive aerosols of different particle sizes in respiratory and alveolar tissues varies significantly. Meanwhile, there were also significant differences in exposure per unit activity concentration, and some reports indicate that for radon progeny of different particle sizes, the effective dose per unit exposure can vary by more than eight times [30]. At the same time, UNSCEAR report pointed that in addition to respiration rate, radon progeny particle size distribution is the factor that has the greatest influence on dose conversions [31]. Therefore, considering the same factor for all aerosols may overestimate or underestimate the converted effective dose value.

The unattached fraction in the tourist mines is in the range of 0.09–0.44 (mean 0.20 ± 0.16) and in the tourist cave is equal to 0.43. The high values of the unattached fraction could be related to the low aerosol concentration in the tourist cave and Show mine 4. The values of this parameter adopted by the ICRP committee for the reference worker are significantly lower compared to the values obtained from direct measurements, and amount to 0.01 for mines and 0.15 for caves.

During the measurements made in the tourist mines, the equilibrium factor F values varied between 0.2 and 0.8 (mean 0.4 ± 0.2), and for the tourist cave, the value was equal to 0.2, which is again significantly different from the values adopted by the ICRP.

The greatest variation was observed between doses determined based on ICRP recommendations and doses calculated from the results of direct measurement of the PAEC, even using the same dose conversions.

## 5. Conclusions

This study made it possible to compare the results of exposure assessments for workers at underground workplaces based on actual conditions prevalent at mining facilities, including the nature of the work performed and direct measurement of agents that affect exposure, such as aerosol size distribution, unattached fraction, and PAEC. The assessments were conducted using either the current regulations in Poland or the International Commission on Radiological Protection (ICRP) recommendations. The results indicated significant limitations in the applicability of ICRP recommendations for assessing effective doses based on measurements of radon activity concentration and standardised dose conversions and equilibrium factors F.

In view of the fact that dose conversions based on real data are much higher than those presently applied (at least in Poland), as well as the latest recommendations of the ICRP [21], examining the aerosol distribution and establishing specific dose conversions for a mine or workplace are advisable, especially in active mines where the radon hazard is significant. The results demonstrate that dose conversions and the proportion of unattached fraction can be estimated from ambient aerosol size distributions [27], even in situations where direct testing of the radioactive aerosol size distribution is not possible. This greatly simplifies radiation exposure monitoring in active underground mines.

In active coal mines, where dust levels are typically elevated, the unattached fraction in determining dose conversions and calculating effective doses may be disregarded. However, in facilities where the air is cleaner, such as tourist mines, the unattached fraction should not be overlooked.

The high variability of the equilibrium factor F also suggests the need to include this parameter in measurements. However, determining the equilibrium factor F necessitates the simultaneous measurement of radon concentration and radon decay product activity concentration (or PAEC). In this situation, for practical reasons, it seems reasonable to solely measure the PAEC and directly calculate the effective dose from it. The results from the measurements demonstrate a strong correlation between the dose assessments based on the PAEC and the calculation according to the ICRP recommendations, with consideration of the actual equilibrium factor F. Regardless of the calculation method employed, the calculated effective doses for workers in tourist facilities are considerably higher than those for workers in active coal mines. This disparity may be attributed to the absence of mechanical ventilation in tourist mines, or if it exists, its low efficacy. Another reason could be the elevated radioactive concentration of nuclides from the uranium series in rocks. However, the measurements of the kerma rate related to external gamma radiation in the show mine where the uranium was mined in the past was at the natural level. In addition, an influx of radon from old inaccessible excavations that have been drilled for hundreds of years by miners cannot be ruled out.

## Figures and Tables

**Figure 1 ijerph-20-05482-f001:**
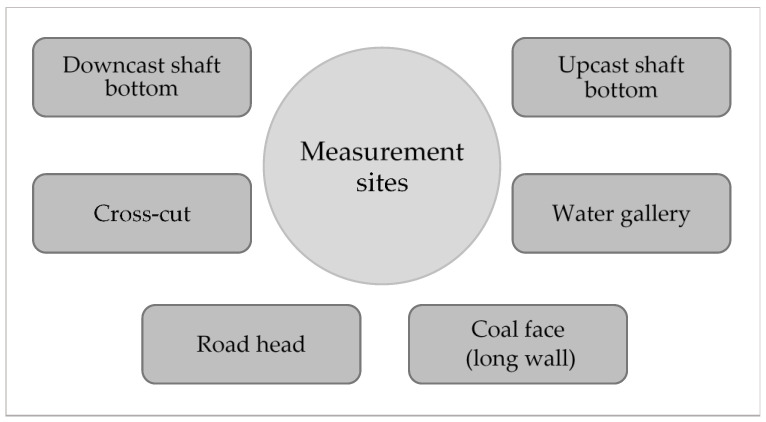
Sites for measuring aerosol concentrations in an active coal mine.

**Figure 2 ijerph-20-05482-f002:**
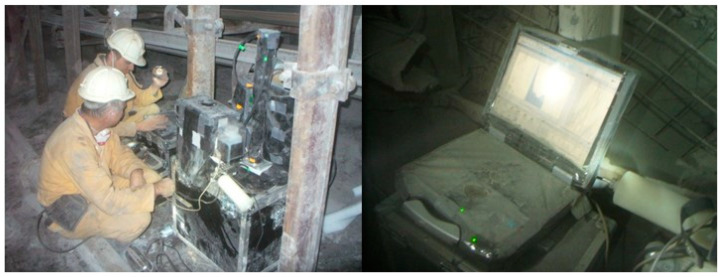
Measurement of the size distribution of ambient aerosol in dusty area of an active coal mine.

**Figure 3 ijerph-20-05482-f003:**
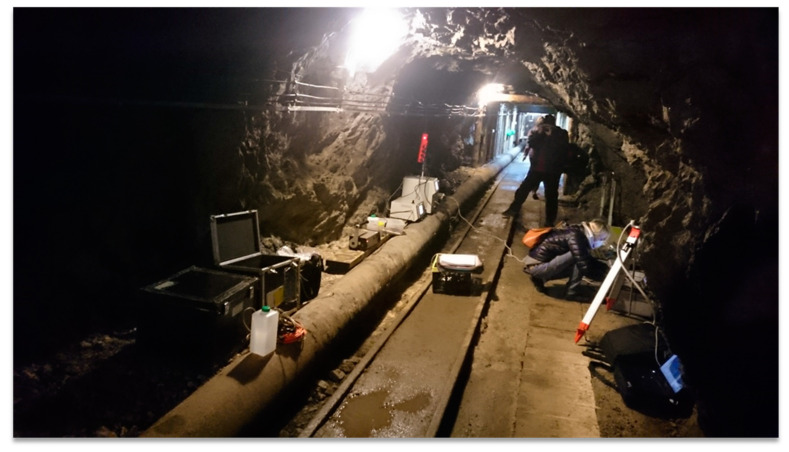
Measurement of the size distribution of ambient aerosol in a show mine.

**Figure 4 ijerph-20-05482-f004:**
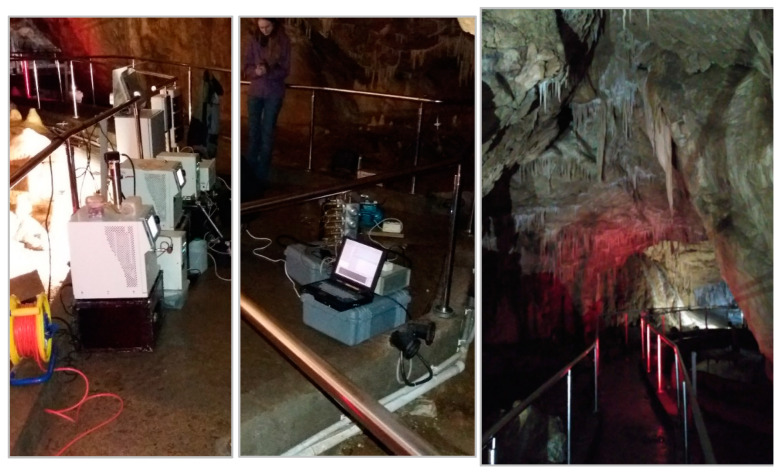
Measurement of the size distribution of ambient aerosol in a tourist cave.

**Figure 5 ijerph-20-05482-f005:**
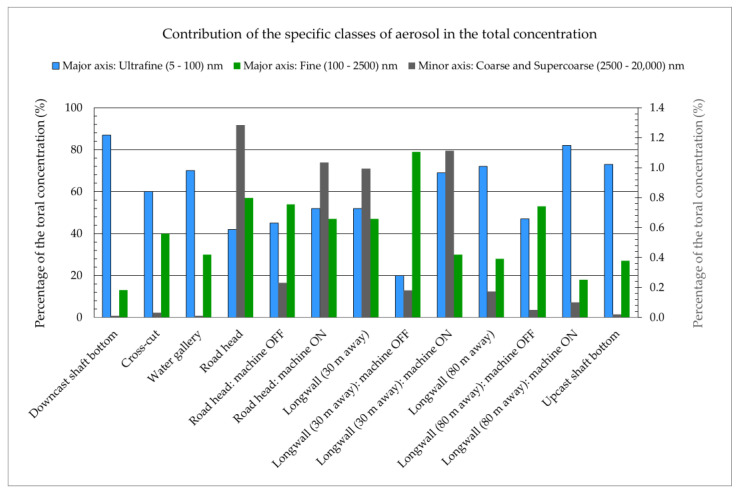
Contribution of the specific classes of aerosol to the total concentration in an active coal mine.

**Figure 6 ijerph-20-05482-f006:**
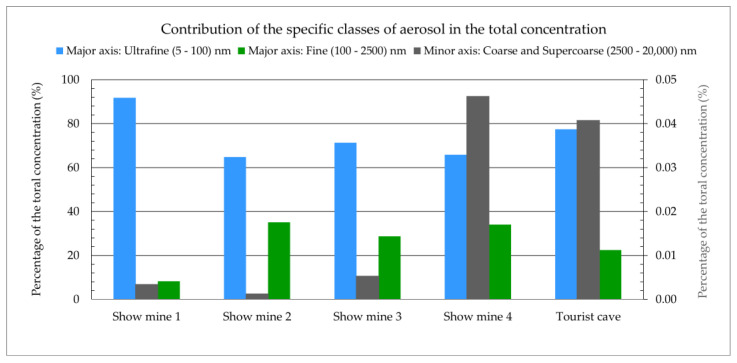
Contribution of the specific classes of aerosol to the total concentration in show mines and a tourist cave.

**Figure 7 ijerph-20-05482-f007:**
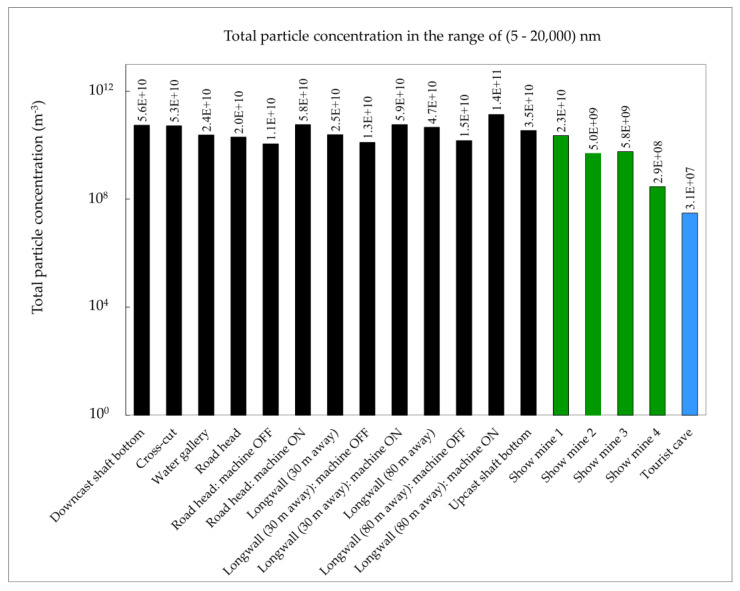
Aerosol concentrations in an active coal mine and in tourist mines.

**Figure 8 ijerph-20-05482-f008:**
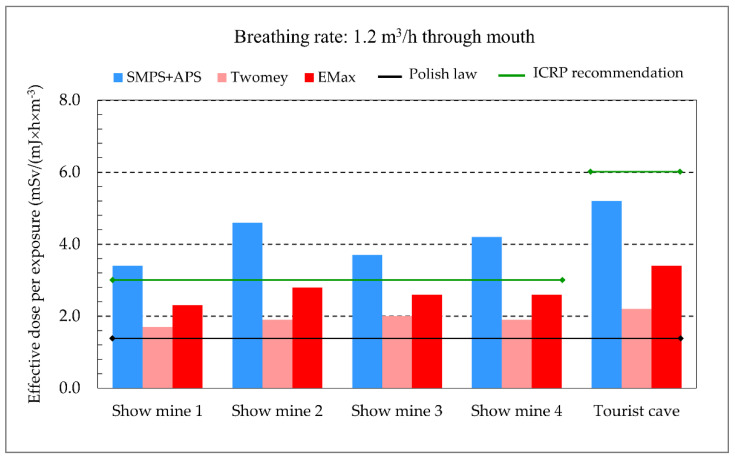
Dose conversions evaluated based on different measurements taken in tourist mines and a cave vs. currently applied or recommended ones.

**Table 1 ijerph-20-05482-t001:** Aerosol concentrations and aerosol size distributions in an active coal mine.

Measurement Place	Total Concentration	Class of Aerosol Particles (μm)		
		LB–0.1	0.1–2.5	2.5–20
	(m^−3^)	%	%	%
Downcast shaft bottom	5.6 × 10^10^	87	13	<0.01
Cross–cut	5.3 × 10^10^	60	40	0.03
Water gallery	2.4 × 10^10^	70	30	0.02
Road head	2.0 × 10^10^	42	57	1.29
Road head: machine OFF	1.1 × 10^10^	45	54	0.23
Road head: machine ON	5.8 × 10^10^	52	47	1.04
Coal face (30 m away)	2.5 × 10^10^	52	47	0.99
Coal face (30 m away): machine OFF	1.3 × 10^10^	20	79	0.18
Coal face (30 m away): machine ON	5.9 × 10^10^	69	30	1.11
Coal face (80 m away)	4.7 × 10^10^	72	28	0.17
Coal face (80 m away): machine OFF	1.5 × 10^10^	47	53	0.05
Coal face (80 m away): machine ON	14.0 × 10^10^	82	18	0.10
Upcast shaft bottom	3.5 × 10^10^	73	27	0.02

Note: LB = 15.1 nm; UB = 17.6 µm except for the downcast shaft bottom, where UB = 19.8 µm.

**Table 2 ijerph-20-05482-t002:** Characteristics of ambient aerosol size distribution in an active coal mine in the size range of 5 nm–20 µm. GM—geometric mean, GSD—geometric standard deviation, d_Av_—average diameter weighted by concentration.

Measurement Place	GM	GSD	d_Av_
	(nm)	-	(nm)
Downcast shaft bottom	52	1.857	66
Cross–cut	80	2.241	111
Water gallery	74	2.004	97
Road head (average)	134	3.128	285
Road head: machine OFF	111	2.254	164
Road head: machine ON	91	3.654	227
Coal face (30 m away)	103	3.053	227
Coal face (30 m away): machine OFF	176	1.954	226
Coal face (30 m away): machine ON	75	2.852	179
Coal face (80 m away)	72	2.175	110
Coal face (80 m away): machine OFF	102	2.177	139
Coal face (80 m away): machine ON	61	1.959	85
Upcast shaft bottom	63	2.114	86

**Table 3 ijerph-20-05482-t003:** Concentration and distribution of aerosols (LB = 3 nm, * LB = 15 nm).

Measurement Place	Total Concentration	Class of Aerosol Particles (μm)		
		LB–0.1	0.1–2.5	2.5–20
	(m^−3^)	%	%	%
Show mine 1	2.3 × 10^10^	92	8	<0.1
Show mine 2	5.0 × 10^9^	65	35	<0.1
Show mine 3	5.8 × 10^9^	71	29	<0.1
Show mine 4 *	2.9 × 10^8^	66	34	<0.1
Tourist cave	4.0 × 10^7^	77	23	<0.1

**Table 4 ijerph-20-05482-t004:** Characteristics of ambient aerosol size distribution in show mines and tourist cave in the size range of 5 nm–20 µm. GM—geometric mean, GSD—geometric standard deviation, d_Av_—average diameter weighted by concentration.

Measurement Place	GM	GSD	d_Av_
	(nm)	-	(nm)
Show mine 1	56	1.862	69
Show mine 2	45	3.246	87
Show mine 3	59	2.515	84
Show mine 4	70	2.177	96
Touristic cave	45	2.855	82

**Table 5 ijerph-20-05482-t005:** Estimated unattached fraction f_p_ and dose conversions at a breathing rate of 1.2 m^3^ × h^−1^ via the mouth, and 0.75 m^3^ × h^−1^ via the nose, with and without taking the unattached fraction into account [27].

Measurement Place	f_p_	Dose Conversion			
		f_p_ = 0		f_p_ ≠ 0	
		Mouth	Nose	Mouth	Nose
		mSv/(mJ × h × m^−3^)			
Downcast shaft bottom	0.01	5.7	4.6	6.0	4.7
Cross-cut	0.01	4.1	3.3	4.4	3.4
Water gallery	0.02	4.3	3.5	4.9	3.7
Road head	0.02	3.6	2.5	4.3	2.8
Coal face (30 m away)	0.02	4.0	2.9	4.6	3.1
Coal face (80 m away)	<0.01	6.9	5.4	7.1	5.5
Upcast shaft bottom	0.01	4.9	4.0	5.3	4.1
Average		4.8	3.7	5.2	3.6

**Table 6 ijerph-20-05482-t006:** Dose conversions calculated from measurements taken in tourist mines and a cave.

Breathing Mode	Method	Show Mine 1	Show Mine 2	Show Mine 3	Show Mine 4	Touristic Cave
		Effective Dose per Exposure mSv/(mJ × h × m^−3^)				
Nose breathing 0.75 m^3^/h	Twomey	1.7	1.9	2.0	1.9	2.2
	Emax	1.9	2.1	2.1	2.0	2.4
	SMPS + APS	2.8	3.7	3.0	3.4	4.1
Mouth breathing 1.20 m^3^/h	Twomey	2.1	2.6	2.6	2.5	3.1
	Emax	2.3	2.8	2.6	2.6	3.4
	SMPS + APS	3.4	4.6	3.7	4.2	5.2
		Unattached fraction f_p_				
		0.09	0.12	0.16	0.44	0.43

**Table 7 ijerph-20-05482-t007:** PAEC in an active coal mine and effective doses calculated: (1)—for annual working time of 1800 h, the dose conversion 1.4 mSv/(mJ × h × m^−3^), and (2)—the dose conversion determined from the aerosol size distributions of ambient aerosols (both for mouth breathing at the rate of 1.2 m^3^ × h^−1^).

Measurement Place	PAEC	Effective Dose (1)	Effective Dose (2)
	(μJ/m^3^)	(mSv)	(mSv)
Cross-cut	0.12 ± 0.02	0.05	0.16
Water gallery	0.66 ± 0.13	1.41	4.94
Road head	0.15 ± 0.03	0.13	0.40
Coal face (30 m away)	0.50 ± 0.10	1.01	3.32
Coal face (80 m away)	0.50 ± 0.10	1.01	5.12
Upcast shaft bottom	0.15 ± 0.03	0.13	0.49

**Table 8 ijerph-20-05482-t008:** Radon activity concentration, PAEC, and equilibrium factor F values in tourist facilities.

Measurement Place	Radon Activity Concentration	PAEC	F
-	(Bq/m^3^)	(μJ/m^3^)	-
Show mine 1	1050 ± 64	4.7 ± 0.6	0.8
Show mine 2	4930 ± 26	12.3 ± 1.5	0.4
Show mine 3	4940 ± 30	13.7 ± 2.5	0.5
Show mine 4	5400 ± 12	6.3 ± 0.3	0.2
Touristic cave	3960 ± 25	4.2 ± 0.2	0.2

**Table 9 ijerph-20-05482-t009:** Radon activity concentration, PAEC, and equilibrium factor F values at various points along the tourist routes in the Show mine 3 and the Show mine 4.

Measurement Site		Radon Concentration	PAEC	F
-		(Bq/m^3^)	(μJ/m^3^)	-
Show mine 3	Place 1	4160 ± 960	1.7 ± 0.4	0.1
	Place 2	2450 ± 560	5.7 ± 0.9	0.4
	Place 3	4430 ± 1020	7.6 ± 0.7	0.3
	Place 4	1060 ± 240	1.9 ± 0.3	0.3
	Average	3030	4.2	0.3
Show mine 4	Place 1	3040 ± 90	4.0 ± 0.8	0.2
	Place 2	810 ± 60	3.3 ± 0.7	0.7
	Place 3	770 ± 50	2.9 ± 0.6	0.6
	Place 4	940 ± 60	2.0 ± 0.4	0.4
	Average	1390	3.0	0.5

**Table 10 ijerph-20-05482-t010:** Comparison of effective doses calculated from measurements made using different methods.

Measurement Site		Values Used for Dose Assessment			
		Dose Conversion k, [mSv/(mJ × h × m^−3^]			
		k = 1.4	k = 3	k = 3	k = 3
		PAEC -	PAEC -	C_Rn_ F = 0.2 [19]	C_Rn_ F: Measured
		Evaluated Effective Annual Dose (mSv)			
Show mine 3	Place 1	4.0	8.6	25.0	12.5
	Place 2	14.1	30.2	14.6	29.2
	Place 3	19.9	40.5	26.6	39.8
	Place 4	4.5	9.7	6.2	9.3
	Average	10.6	22.3	18.1	22.7
Show mine 4	Place 1	9.8	21.1	18.2	18.2
	Place 2	8.1	17.3	4.7	16.3
	Place 3	7.1	15.1	4.4	13.3
	Place 4	4.8	10.3	5.5	10.9
	Average	7.5	15.6	8.2	14.7

## Data Availability

All the data are available on request.

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
