# Peer review of "Influence of Dose Conversions, Equilibrium Factors, and Unattached Fractions on Radon Risk Assessment in Operating and Show Underground Mines"

_ijerph, 2023, doi:10.3390/ijerph20085482_

Round 1

Reviewer 1 Report

The presented article is devoted to the assessment of radiation doses of miners and tourists in coal mines. The aerosol size distributions of ambient aerosols at key workplaces and the distributions of radioactive aerosols containing radon decay products were determined. Based on these studies, dose coefficients used for dose assessment and unattached fraction contributions were determined. The authors managed to get unique and very important results. This is especially important considering how difficult and risky it is to take measurements in deep operating mines. I recommend it for publication with the minor corrections. 

Some comments listed below.

#1. In my opinion, the title of the article could be somewhat corrected. As I understand, the measurements were carried out only in one mine at different horizons at different key points. On the one hand, the results of the study in one mine can hardly be extended to all the Polish mines.  On the other hand, the patterns obtained in this work may be valid for coal mines located not only in Poland, but also in other countries. In this regard, in the title of the article it is better either to indicate the specific area where the measurements were carried out (not the Polish mines in general), or not to give a reference to the terrain at all, e.g. "Radon exposure doses assessment in operating and show underground mines".

#2. In the caption to Fig 1, it seems to me it is woud pointing out that the decay schemes are presented in an abbreviated form - the members of the decay series between Uranium-238 and radium-226 (thorium-232 and radium-224) are omitted.

#3. In formula 2, please add F=...

#4. In section 2, "Materials and methods", it would be good to clarify where the studied mines and cave are located. Maybe show it on the map. If this is impossible, for example, for reasons of commercial secrecy, then at least to mentioned the area (e.g. Upper Silesia). In addition, if possible, it would be good to give a scheme of the operating and show mines.

#5. In Fig. 2, the measurement sites do not completely correspond to the description on lines 241-247. The "Downcast shaft" and the "Road head" are missing in the text.

#6. In Figures 6 and 7, the right ordinate axis is not signed. Apparently, the fraction content of 2500 - 20,000 nm is shown along this axis. However, it needs to be mentioned somewhere.

#7. In lines 524-527 you write: "The calculated effective doses to which workers in tourist facilities may be exposed, irrespective of the calculation method adopted, are significantly higher than the doses to workers in active coal mines. This may be due to the lack of mechanical venutilization in tourist mines or, if it existed, its low efficiency." However, as you mentioned in line 259, in tourist mines uranium ores were mined in the past. Could higher values of CRn and PAEС in tourist mines be associated with increased uranium content in rocks and, correspondingly, high radon exhalation from the walls of mines, and not only with the ventilation regime?

Author Response

Reviewer 1

Comments and Suggestions for Authors

The presented article is devoted to the assessment of radiation doses of miners and tourists in coal mines. The aerosol size distributions of ambient aerosols at key workplaces and the distributions of radioactive aerosols containing radon decay products were determined. Based on these studies, dose coefficients used for dose assessment and unattached fraction contributions were determined. The authors managed to get unique and very important results. This is especially important considering how difficult and risky it is to take measurements in deep operating mines. I recommend it for publication with the minor corrections. 

Some comments listed below.

#1. In my opinion, the title of the article could be somewhat corrected. As I understand, the measurements were carried out only in one mine at different horizons at different key points. On the one hand, the results of the study in one mine can hardly be extended to all the Polish mines.  On the other hand, the patterns obtained in this work may be valid for coal mines located not only in Poland, but also in other countries. In this regard, in the title of the article it is better either to indicate the specific area where the measurements were carried out (not the Polish mines in general), or not to give a reference to the terrain at all, e.g. "Radon exposure doses assessment in operating and show underground mines".

Answ. OK. We changed the title according to your suggestions as well as other reviewers' comments.

#2. In the caption to Fig 1, it seems to me it is would pointing out that the decay schemes are presented in an abbreviated form - the members of the decay series between Uranium-238 and radium-226 (thorium-232 and radium-224) are omitted.

Answ. This is a valid point, but one of the reviewers wanted to remove this drawing because it is basically known.

#3. In formula 2, please add F=...

Answ. We added this symbol to the equation.

#4. In section 2, "Materials and methods", it would be good to clarify where the studied mines and cave are located. Maybe show it on the map. If this is impossible, for example, for reasons of commercial secrecy, then at least to mention the area (e.g. Upper Silesia). In addition, if possible, it would be good to give a scheme of the operating and show mines.

Answ. We added such panel at the beginning of the section 3.

#5. In Fig. 2, the measurement sites do not completely correspond to the description on lines 241-247. The "Downcast shaft" and the "Road head" are missing in the text.

Answ. Thank’s a lot. The description was partially misleading. We have corrected the items.

#6. In Figures 6 and 7, the right ordinate axis is not signed. Apparently, the fraction content of 2500 - 20,000 nm is shown along this axis. However, it needs to be mentioned somewhere.

Answ. The figures (now Figure 5 and Figure 6) were corrected according to your recommendation.

#7. In lines 524-527 you write: "The calculated effective doses to which workers in tourist facilities may be exposed, irrespective of the calculation method adopted, are significantly higher than the doses to workers in active coal mines. This may be due to the lack of mechanical ventilation in tourist mines or, if it existed, its low efficiency." However, as you mentioned in line 259, in tourist mines uranium ores were mined in the past. Could higher values of CRn and PAEС in tourist mines be associated with increased uranium content in rocks and, correspondingly, high radon exhalation from the walls of mines, and not only with the ventilation regime?

Answ. I hope, we will discuss this question in other paper. Any way in one of these objects we have performed very detailed measurements, and the main reason of the enhanced concentration of radon was poor ventilation caused by temporary and regular switching off ventilation. Unfortunately, we didn’t measure the radon exhalation rate from the walls but we made measurements of kerma rate due to external gamma radiation in the object where the uranium was mined in the past. Strange but its value was normal. Another reason could be the inflow of radon from unavailable excavations, which had been hollowed out by miners for hundreds of years. Anyway, we added some new possibilities to the paper.

The manuscript has been reviewed and corrected by a native speaker.

The changes according to reviewer’s recommendations are marked in yellow.

Thank you for the review.

Best regards,

Authors

Author Response

Reviewer 2

In this paper, the authors have done many measurements on aerosol concentrations and radon concentrations in multiple, and analyzed for annual effective dose levels at the site according the radon concentration level.It is recommended to publish after minor revision. So, the paper needs improvement before acceptance for publication. My detailed comments are as follows:

  1. Too many keywords are extracted. For example, underground active mines and Show mines can be combined into a keyword. The choice of keywords should be showing the most important contents of this article so that the reader can focus on these information.

Answ. The keywords were changed.

  1. The specific content mentioned in the title of the article is radon hazard, but this paper only introduces and explores the relationship between aerosol and radon progeny, rather than radon hazard, It is suggested to modify the title or analyze and discuss the harm of radon more.

Answ. We have changed the title according to your suggestion.

  1. Figure 1 is well known, should be deleted.

Answ. The figure was removed.

  1. The research had found that the deposition efficiency of radioactive aerosols of different particle sizes in respiratory and alveolar tissues varies significantly. Meanwhile, there were also significant differences in exposure per unit activity concentration. CRP137 indicates that for radon progeny of different particle sizes, the effective dose per unit exposure can vary by more than 8 times[1].At the same time, UNSCEAR's 2019 report pointed that in addition to respiration rate, radon progeny particle size distribution is the factor that has the greatest influence on radon exposure dose conversion coefficient[2]. Therefore, considering the same factor to all aerosols may oversize the converted effective dose value.

[1]   ICRP P.Occupational intakes of radionuclides,part 1[Z].Publication 130 ann ICRP,2015.

[2] HARRISON D. Lung cancer risk and effective dose coefficients for radon. Journal of Radiological Protection,.2021, 41(2), pp. 433-441.

Answ. We have considered the items.

  1. In page 15 lines 387 to 390, it is mentioned that ‘It is also important to note the relatively constant ratio of the values of the dose coefficients calculated when taking into account the ambient aerosol size distribution to the values of these factors calculated for results obtained by RPPSS spectrometer. ’However, there is a column of abnormal data according the providing data which disagrees with the relatively constant ratio in this sentence. Which is shown in Table 1with red marked. Please explain the reason for this abnormal data. Meanwhile, the accuracy of the relatively constant ratio remains to be verified in that the obtained data are less, so, it is recommended to add some data to support this conclusion.

Table 1 Show mine 1

Show mine 4

Show mine 2

Show mine 3

Touristic cave

Twomey

1.7

1.9

1.9

2

2.2

Emax

1.9

2

2.1

2.1

2.4

SMPS+APS

2.8

3.4

3.7

3

4.1

Twomey

2.1

2.5

2.6

2.6

3.1

Emax

2.3

2.6

2.8

2.6

3.4

SMPS+APS

3.4

4.2

4.6

3.7

5.2

Answ. The calculated ratios are as follows:

Indeed, the indicated value differs the most from the others. But it was more about the trend. Perhaps, however, it was too bold a statement, so we removed this sentence.

The manuscript has been reviewed and corrected by a native speaker.

The changes according to reviewer’s recommendations are marked in yellow.

Thank you for the review.

Best regards,

Authors

Reviewer 3 Report

Two small remarks from my side: 

1. Line 222-224: "The potential alpha energy values obtained from each stage provide input to the two independent deconvolution algorithms Twomey [25] and EMax (Expectation Maximisation) [26]" Please add breif information about deconvolution algorithms paying atention on their key diferences.

2. Line 211: "determining DCF dose coefficients". Please explain "DCF". If DCF means "Dose Conversion Factor", please remove "dose coefficients"

Author Response

Reviewer 3

  1. Line 222-224: "The potential alpha energy values obtained from each stage provide input to the two independent deconvolution algorithms Twomey [25] and EMax (Expectation Maximisation) [26]" Please add brief information about deconvolution algorithms paying attention on their key differences.

Answ. We have added some information about these algorithms to the paper.

  1. Line 211: "determining DCF dose coefficients". Please explain "DCF". If DCF means "Dose Conversion Factor", please remove "dose coefficients."

Answ. We have introduced changes. Now the only term is “Dose conversion” as denoted in the ICRP report no. 137.

The manuscript has been reviewed and corrected by a native speaker.

The changes according to reviewer’s recommendations are marked in yellow.

Thank you for the review.

Best regards,

Authors
